# The Effects of Dietary Supplements, Nutraceutical Agents, and Physical Exercise on Myostatin Levels: Hope or Hype?

**DOI:** 10.3390/metabo12111146

**Published:** 2022-11-20

**Authors:** Heitor O. Santos, Henrique S. Cerqueira, Grant M. Tinsley

**Affiliations:** 1School of Medicine, Federal University of Uberlandia (UFU), Uberlandia 38408-100, Brazil; 2School of Medicine, University of São Paulo (USP), Ribeirão Preto 14049-900, Brazil; 3Department of Kinesiology and Sport Management, Texas Tech University, Lubbock, TX 79409, USA

**Keywords:** myostatin, muscle mass, supplements, whey protein, amino acids

## Abstract

Myostatin, a secreted growth factor belonging to the transforming growth factor β (TGF-β) family, performs a role in hindering muscle growth by inhibiting protein kinase B (Akt) phosphorylation and the associated activation of hypertrophy pathways (e.g., IGF-1/PI3K/Akt/mTOR pathway). In addition to pharmacological agents, some supplements and nutraceutical agents have demonstrated modulatory effects on myostatin levels; however, the clinical magnitude must be appraised with skepticism before translating the mechanistic effects into muscle hypertrophy outcomes. Here, we review the effects of dietary supplements, nutraceutical agents, and physical exercise on myostatin levels, addressing the promise and pitfalls of relevant randomized clinical trials (RCTs) to draw clinical conclusions. RCTs involving both clinical and sports populations were considered, along with wasting muscle disorders (e.g., sarcopenia) and resistance training-induced muscle hypertrophy, irrespective of disease status. Animal models were considered only to expand the mechanisms of action, and observational data were consulted to elucidate potential cutoff values. Collectively, the effects of dietary supplements, nutraceutical agents, and physical exercise on myostatin mRNA expression in skeletal muscle and serum myostatin levels are not uniform, and there may be reductions, increases, or neutral effects. Large amounts of research using resistance protocols shows that supplements or functional foods do not clearly outperform placebo for modulating myostatin levels. Thus, despite some biological hope in using supplements or certain functional foods to decrease myostatin levels, caution must be exercised not to propagate the hope of the food supplement market, select health professionals, and laypeople.

## 1. Introduction

Myostatin, also known as growth differentiation factor 8, is a transforming growth factor-β family member that negatively regulates skeletal muscle growth [1]. Myostatin genetic blockade displays an intense and generalized accretion in skeletal muscle mass, as shown in animal models [2,3,4]. In humans, myostatin is also involved in muscle homeostasis as its expression is regulated during muscle atrophy, and hence myostatin has gained interest in the management of muscle wasting disorders in adults [2].

Since myostatin was cloned in 1997 [5], many myostatin-blocking agents have gained attention in agriculture applications and in the management of muscle diseases and disorders [6], e.g., injuries, sarcopenia, wasting/cachexia, Duchenne type muscular dystrophy, Becker muscular dystrophy, facioscapulohumeral muscular dystrophy, etc. [7,8]. In addition, experimental research has shed light on myostatin inhibition in muscle to improve insulin resistance by enhancing glucose uptake [9,10,11]. Despite emerging myostatin-blocking drugs, myostatin alone is apparently not classified by guidelines as a therapeutic target for muscle diseases in the same manner as low-density lipoprotein-cholesterol for hypercholesterolemia and related heart diseases, blood pressure for hypertension, fasting blood glucose for diabetes, or blood total testosterone for male hypogonadism [12,13,14,15,16,17,18].

To date, there are ongoing randomized clinical trials (RCTs) investigating the effects of myostatin-blocking drugs [19,20,21]. Interestingly, many RCTs focusing on nutritional supplementation have recently assessed myostatin levels as part of biochemistry screening [22,23,24], but a critical review is imperative to unify these findings and discuss the clinical implications of changing myostatin levels through nutritional supplementation.

This review primarily discusses the effects of nutritional supplementation and nutraceutical agents on myostatin levels in humans. In addition, we discuss the mechanisms, proposed laboratory ranges in different populations, and changes in myostatin levels in response to physical exercise, particularly resistance training (RT), to expand the non-pharmacological landscape. Pharmacological agents are briefly discussed to portray an indirect comparison with non-pharmacological strategies.

## 2. Mechanisms

In addition to embryogenesis, myostatin is expressed and secreted by skeletal muscle in adulthood and thus performs a role in suppressing muscle hypertrophy in different populations [25]. Myostatin is a potent negative regulator of satellite cell activation and self-renewal, and upregulates ubiquitin-associated genes such as atrogin-1, muscle RING-finger protein-1 (MuRF-1), and 14-kDa ubiquitin-conjugating enzyme E2 [25,26]. Myostatin is released into the circulation and acts systemically by binding to cell-surface receptors. Concerning myostatin receptors, it has a high affinity for the activin type IIB receptor and weak affinity for the activin receptor type IIA, inducing muscle wasting by acting on multiple systems [25].

More specifically, myostatin leads to receptor-mediated phosphorylation of Smads 2 and 3, in which myostatin-mediated Smad signaling is activated by binding of the mature myostatin peptide to plasma membrane-associated activin type IIB and type IIA receptors [27]. Indeed, myostatin-mediated Smad signaling is a decisive pathway in declining myofibrillar protein synthesis while increasing protein degradation, such as Smads 2 and 3 inhibit the protein kinase B/mammalian target of rapamycin (Akt/mTOR) signaling and trigger the expression of ubiquitin proteasome E3 ligases, atrogin-1, and MuRF-1 [27,28]. This inhibition of Akt phosphorylation is associated with a negative cascade of effects in hypertrophic pathways by decreasing the actions of insulin-like growth factor-1 (IGF-1) and phosphatidylinositol 3-kinase/protein kinase B (PI3K), and increased production of the transcription factor FoxO1, a key stimulator of atrogin-1 and other atrophy-related genes (atrogenes) [29].

Accordingly, this mechanistic background establishes the interest in blocking myostatin (or the activin type IIB receptor) pharmacologically, and attempting to lower myostatin levels by nature-based intervention as a means of alleviating skeletal muscle loss.

## 3. Laboratory Levels

Serum myostatin levels must be discussed considering different countries, populations, age groups, and laboratory methods in order to portray a clinical rationale for their alterations.

Seemingly, the largest investigation assessing myostatin levels was a prospective cohort study conducted in France [30]. Serum myostatin levels ranged from ~27 to ~33 ng/dL among 1121 young and older males [30], in whom serum myostatin increased slightly with age until 57 years (r = −0.07; 0.09 standard deviation per decade; *p* < 0.05), then declined. It is imperative to note that the myostatin levels from this study were sharply higher than the studies below [31,32,33]. Such a discrepancy could be happened because the researchers used nonfasting serum collection (1300 h) or because of the method (ELISA, Immundiagnostik AG, Bensheim, Germany).

The second largest study that examined myostatin levels included a Korean cohort of a similar sample size to the above study (n = 1053 individuals ≥ 70 years; 519 men and 534 women). Serum myostatin levels were 3.7 ± 1.2 ng/mL for men and 3.2 ± 1.1 ng/mL for women (*p* < 0.001 between groups), using a sandwich enzyme immunoassay kit (R&D Systems, Inc., Minneapolis, MN, USA) [31]. Although a certain decline in serum myostatin levels can be expected for older subjects, seemingly due to physiological actions to partially counteract age-related muscle wasting [34], in this Korean study [31], mean serum myostatin levels for men were approximately nine times lower than the result observed for men of the similar age group in the France cohort (28.4 ± 12.3 ng/mL for >70 to 80 yr, n = 345) [30]. A smaller American study detected 8.0 ± 0.3 ng/mL and 7.0 ± 0.4 ng/mL serum myostatin levels, using a personalized ELISA method, for young (18–35 years, n = 50) and older men (60–75 years, n = 58), respectively, in which myostatin levels were significantly higher in young men (*p* = 0.03, between group) [32]. Mean serum myostatin levels of both young and older men were approximately four times lower than the results for subjects of a similar age group in the French cohort (30.5 ± 9.5 ng/mL for 20 to 30 years, n = 76; 28.4 ± 12.3 ng/mL for >70 to 80 years, n = 345) [30].

The American study did not observe significant differences between menstruating women (n = 33; 7.0 ± 2.7 ng/dL) and naturally menopausal (n = 24; 6.7 ± 2.8 ng/mL) and surgically menopausal women (n = 37; 6.7 ± 2.7 ng/mL) [32], and the values were clinically close to those from a Polish study including Caucasian women who were perimenopause or postmenopausal (n = 300), in which serum myostatin levels were 6.58 ± 3.59 ng/mL (Human Myostatin ELISA Kit, SunRed Biotechnology Company) [33].

### General Considerations

When considering appropriate reference values for healthy populations, we lend greater weight to the values from studies that are most similar to each other [31,32,33], whereas the high discrepancy between the French cohort values [30] and the other studies make its integration into reference values challenging. Taken together, serum myostatin levels from different populations are displayed in Table 1, not only for different age groups and sexes of healthy individuals, but also for various ailments such as kidney, heart, lung, muscular and skin diseases, diabetes, and metabolic syndrome [30,31,32,33,35,36,37,38,39,40,41,42,43,44,45,46,47,48,49]. However, further research is crucial to propose specific reference ranges for circulating myostatin levels.

## 4. Functional Foods and Supplements

### 4.1. Proteins, Amino Acids, and Derivatives

Adequate protein intake is fundamental to optimize muscle hypertrophy and minimize muscle catabolism, in which essential amino acids, e.g., branched-chain amino acids (BCAAs), increase mTOR phosphorylation, the sequential activation of 70-kD S6 protein kinase, and the eukaryotic initiation factor 4E-binding protein 1 [50].

Interestingly, amino acids and proteins are the most studied nutrients in regard to the possible modulation of myostatin [23,51,52,53,54,55,56,57,58,59], particularly through stimulating the production of IGF-1, an anabolic hormone that suppresses myostatin signaling pathways [60,61]. In this subtopic, we discuss the effect of dietary proteins, protein supplements, and supplementation of amino acids and their derivatives on myostatin levels.

#### 4.1.1. Egg

Although both egg yolk and egg whites are rich in protein, the yolk comprises of a plethora of nonprotein nutrients with putative anabolic properties (e.g., vitamins, minerals, microRNAs, lipids, phosphatidic acid, and other phospholipids) [52,62,63].

In a 12-week RT intervention in young men (n = 30), consumption of whole eggs (3 units) or isonitrogenous consumption of egg whites (6 units) reduced serum myostatin levels (–0.1 ng/mL and –0.06 ng/mL for egg whites and whole eggs, respectively), with no significant differences between groups. However, the variation is clinically negligible despite statistical significance [52].

#### 4.1.2. Milk

Cow milk contains a food matrix consisting of minerals, fats, lactose, water, and proteins (80% casein and 20% whey protein), with the majority of these nutrients viewed as favorable for muscle accretion [60].

In resistance-trained young males (n = 30; RT experience: 15 ± 2 months) who underwent 6 weeks of linear periodized RT (4 times/week), high-protein milk (156 kcal; 30 g of protein, 6 g whey, 24 g fat, and 10 g carbohydrate) ingested immediately postexercise reduced serum myostatin levels compared to placebo (*p* < 0.05) [24]. However, despite the isoenergetic drink in the placebo group, caloric intake was higher in the milk group compared to the placebo group (2909.2 ± 109.2 and 2653.3 ± 109.1 kcal, respectively); additionally, higher protein intake was also observed in the milk group compared to placebo (2.3 vs. 1.4 g/kg body weight/d).

In an acute crossover study of healthy resistance-trained males (n = 7), myostatin mRNA expression from vastus lateralis did not change 3 h after 90 min of power resistance exercise with a postexercise meal rich in milk protein (600 mL chocolate milk and 85 g muesli bar; 102 g carbohydrate, 34 g protein, and 22 g fat), power resistance exercise without postexercise meal and rest (i.e., no exercise and postexercise meal) [64].

#### 4.1.3. Whey Protein

Whey protein, one of the two major protein groups of cow milk, is one of the best proteins for stimulating muscle protein synthesis rate and muscle growth thanks to its high essential amino acid content [65,66]. However, to date, its influence on myostatin expression has not been fully elucidated.

Hulmi et al. (2008) did not observe acute changes in myostatin mRNA from vastus lateralis expression under whey protein isolate supplementation (15 g of whey protein immediately after exercise) combined with an RT bout (leg press, 5 × 10 repetitions) in trained middle-aged to older men (n = 9) [67]. However, myostatin mRNA levels statistically decreased at the 48 h assessment (*p* = 0.03) and demonstrated a trend (*p* = 0.06) for reduction at 1 h postexercise for placebo (absolute levels were not reported).

In a later study including untrained young men ingesting double the dose of whey protein isolate (15 g of whey protein both before and after exercise) as compared to the previous study, Hulmi et al. (2009) observed neither acute (1 and 48 h after RT) nor chronic (after 21 weeks post-RT) changes in myostatin mRNA expression from vastus lateralis under whey protein isolate supplementation along with the leg press protocol (5 × 10 repetitions) [68]. Similar to their previous study, myostatin mRNA expression changed only in the placebo group, at 1 h post-exercise, in which there was a 31% drop (*p* < 0.02).

Paoli et al. (2015) analyzed the effects of high protein intake (1.8 g protein/kg body weight/d) provided by whey protein supplementation (15–20 g of protein during warm-up and 1 h after RT) in addition to the habitual diet versus normal protein intake (0.85 g protein/kg body weight/d) on plasma myostatin levels in active young men without RT experience (n = 18), who underwent 8 weeks of RT [22]. Taking into account the plasma myostatin measurements pre and post the last RT session, it increased in the high protein group (pre-training session 3.66 ± 1.42 ng/mL, post-training session 12.0 ± 2.5 ng/mL; *p* = 0.02), while no change was detectable in the normal protein group (pre-training session 4.23 ± 2.59 ng/mL, post-training session 3.64 ± 2.09 ng/mL; *p* > 0.05). As the authors state, this response is paradoxical because a decline in myostatin levels would be expected in the group with higher protein intake. However, values greater than triple the values of the normal protein group were observed.

In a study of healthy older adults who underwent five days of immobilization of one knee with a full-leg cast, both a whey protein nutritional supplement (n = 11; 20.7 g protein, 9.3 g carbohydrate, and 3.0 g fat) twice daily or control (n = 12; no supplementation) increased the relative myostatin mRNA expression from vastus lateralis (*p* < 0.05; absolute values were not reported) with no differences between groups [58]. Furthermore, atrophy and decreased leg strength occurred in both groups, despite protein consumption being 45% higher in the supplementation group than in the control group (125 ± 6 g/d vs. 86 ± 4 g/d, which represents ~1.6 and ~1.1 g/kg/body weight, respectively).

In a crossover study of untrained college-aged men (n = 10) submitted to supplementation with whey protein isolate (25 g), maltodextrin (25 g), or placebo before to RT sessions (3 × 10 at 80% 1 RM for bilateral hack squat, leg press, and leg extension), myostatin mRNA expression from vastus lateralis 6 h following exercise was significantly reduced in all conditions (−29.4% in placebo, −24.7% in protein, and −3.4% in carbohydrate) [69].

Whey protein supplementation mixed with carbohydrate after RT (knee extension 3 times/week for 8 weeks) appears to have an acute effect on myostatin expression in patients with chronic obstructive pulmonary disease, but, chronically, the levels return to the baseline status. In one investigation, patients were divided into two groups: whey protein concentrate supplementation mixed with glucose polymer carbohydrate (19 g protein, 49 g carbohydrates; n = 27) or placebo (non-caloric beverage; n = 32); a healthy control group (n = 21) also received placebo, and all groups underwent RT. Myostatin mRNA expression from vastus lateralis was significantly reduced at 24 h (*p* < 0.05; absolute values are not shown) in all groups, but was restored to the baseline status at 4 and 8 weeks in all groups [59].

Recently, a study of hospitalized older patients (n = 41) showed that supplementation of leucine-enriched whey protein (20 g of whey + 3 g of post-workout leucine) and placebo, in conjunction with 12 weeks of RT, did not alter serum myostatin levels [23].

#### 4.1.4. Amino Acids

In a study consisting of young men (n = 41) performing a single RT session (four sets of 10 repetitions of leg press and leg extension at 80% 1 RM) with peri-exercise supplementation of carbohydrates (1.5 g/kg), carbohydrates plus BCAA (120 mg/kg BCAA), carbohydrates plus leucine (120 mg/kg), or placebo, the mean myostatin expression in vastus lateralis over 360 min postexercise was higher for those who supplemented with carbohydrates (1.00 ± 0.09) and carbohydrates plus BCAA (1.05 ± 0.08) than carbohydrates plus leucine (0.92 ± 0.07) or placebo (0.90 ± 0.05) [51]. This result is biologically intriguing, as a reduction in myostatin would be expected with both BCAA and leucine supplementation.

In a study of postmenopausal women undergoing 8 weeks of RT (n = 20), both 9 g BCAA supplementation (half consumed 30 min before the training session and the other half consumed within 30 min of the end of the session) and placebo reduced serum myostatin concentrations (−0.7 ng/mL for BCAA and −0.4 ng/mL for placebo) with no difference between groups [53].

In patients recovering from hip replacement (n = 20) and underdoing to an 8 week rehabilitation program, amino acid supplementation (two sachets of 4 g daily: 1250 mg of l-leucine, 650 mg of l-lysine; 625 mg of l-isoleucine, 625 mg of l-valine, 350 mg of l-threonine, 150 mg of l-cystine, 150 mg of l-histidine, 100 mg of l-phenylalanine, 50 mg of l-methionine, 30 mg of l-tyrosine, 20 mg of l-tryptophan; 0.15 mg of vitamin B6, and 0.15 mg of vitamin B1) significantly reduced serum myostatin levels and for placebo (1.2 ± 0.2 vs. 0.9 ± 0.3 ng/mL; *p* = 0.01; 1.3 ± 0.3 vs. 1.1 ± 0.4 ng/mL; *p* = 0.03 for amino acids and placebo, respectively), with no differences between groups [70]. However, when considering only patients with sarcopenia, only the supplemented group showed a significant decrease in myostatin levels (1.3 ± 0.3 vs. 0.9 ± 0.5 ng/mL; *p* = 0.04), while the placebo group did not (1.2 ± 0.3 vs. 0.9 ± 0.5 vs. 1.0 ± 0.7 ng/mL; *p* = 0.12).

In a study that provided 15 g of BCAA (7.5 g of leucine, 3.75 of isoleucine, and 3.75 of valine) to stable patients with alcoholic cirrhosis (n = 6) and healthy controls (n = 8), myostatin mRNA levels from vastus lateralis were higher at baseline for the patients with alcoholic cirrhosis than the control group (*p* < 0.001; absolute values were not reported), but none of the groups showed a significant change after 7 h of supplementation [56].

#### 4.1.5. HMB

Beta-hydroxy-beta-methylbutyrate (HMB) is a leucine metabolite with anti-catabolic and anabolic properties. Instead of leucine supplementation per se, HMB supplementation gained attention because only 5% of leucine is metabolized into HMB, and it would be necessary to ingest 60 g of leucine to obtain 3 g of HMB—the most common dosage used [71].

In patients with bronchiectasis (n = 28), HMB-enriched protein supplementation (330 kcal, 18 g protein, 1.5 g HMB, and 1.7 g fructooligosaccharide, Ensure Plus Advance^®^) along with 12 week pulmonary rehabilitation improved body composition and muscle strength, whereas there was a trend (*p* = 0.06) for a reduction in plasma myostatin levels similar to those who received placebo plus pulmonary rehabilitation (3 ± 1.5 to 2.42 and 2.55 ± 1 to 2.43 ± 0.75 ng/mL for supplementation and placebo, respectively) [57].

#### 4.1.6. Creatine

Creatine supplementation is a common nutritional strategy to augment increases in muscle strength and lean body mass particularly due to intramuscular water retention [72,73].

In an acute study, young men (n = 9) were randomized to creatine supplementation (21 g/d, 3 doses of 7 g) or placebo (maltodextrin) for 5 days, then followed one bout of RT (10 × 10 repetitions of one-leg extension at 80% 1 repetition maximum). Myostatin mRNA expression from vastus lateralis was reduced by 35% 24 h after the exercise regardless of creatine supplementation, however, it returned to the baseline status result after 72 h [55].

In an 8 week RT intervention (upper and lower-limb exercises 3 x/week) in healthy men (n = 24) who received creatine supplementation (0.3 g/kg/d body weight at week 1 and 0.05 g/kg/d body weight/d for the remainder of the intervention) or placebo, serum myostatin levels decreased in both groups at weeks 4 and 8 (*p* < 0.05; absolute values were not reported), but there was a more pronounced decrease in the group receiving creatine [54].

### 4.2. Non-Protein Supplements

In this subtopic, we discuss the effect of functional foods, herbal extracts, and supplementation with vitamins and nonessential antioxidants. Although certain functional foods contain some amount of protein (i.e., cocoa and spirulina), their protein and amino acid content are less relevant as compared to the sources in the previous subtopic.

#### 4.2.1. Brown Seaweed

Brown seaweed is rich in bioactive compounds, especially unique polyphenols (e.g., phlorotannins) [74]. Seemingly, Willoughby et al. (2004) were the first researchers to investigate the effects of a supplement on myostatin concentrations, who used a commercial supplement based on brown algae (*Cystoseira canariensis*) [75]. The researchers subjected 22 men with no previous strength training experience to 12 weeks of RT, who received 1200 mg/d of brown seaweed or placebo, and serum myostatin concentrations increased equivalently between the groups.

#### 4.2.2. Spirulina

Spirulina is a commercially available cyanobacterial biomass extract and comprises essential fats (e.g., gamma-linolenic oleic acids), vitamins (mainly B12), provitamin A (i.e., β-carotenes), and minerals (mainly iron, calcium, and phosphorous), and is easily digested due to the lack of cellulose cell walls [76,77,78]. In a study that analyzed 40 wrestlers who underwent a 12 day gradual weight loss protocol, and were divided into two groups: spirulina (3 g/d) or placebo, there was a significant decrease in serum myostatin levels only in the SP group (−0.1 ng/mL, *p* = 0.005); [79]; however, despite the statistical significance, this magnitude of decrease is of doubtful clinical importance.

#### 4.2.3. Cocoa

Cocoa powder is a source of macronutrients (fats, carbohydrates, and proteins), dietary fiber, magnesium, potassium and caffeine, but the polyphenols, particularly those of the flavonoid class, are its main functional nutrient [80]. In trained endurance athletes (n = 44) undergoing a 10 week endurance training intervention, neither daily supplementation with 5 g of flavonoid-rich cocoa powder (425 mg total) nor placebo (5 g of maltodextrin) changed serum myostatin levels over time [81].

#### 4.2.4. Epicatechins

Molecularly, epicatechins can favor the muscle structure, function, metabolism, and growth due to their antioxidant and anti-inflammatory properties and stimulation of mitochondrial biogenesis and signaling proteins [82].

Older patients with sarcopenia (n = 62) were randomized to RT, epicatechin (1 mg/kg/d), RT plus epicatechin (1 mg/kg/d) or placebo for 8 weeks [83]. Older patients with sarcopenia (n = 62) were randomized to RT, epicatechin (1 mg/kg/d), RT plus epicatechin (1 mg/kg/d), or placebo for 8 weeks. Plasma myostatin levels were significantly reduced only for the RT groups. There was an ~49% greater reduction in the RT plus epicatechin group than in the RT group, but there was no statistically significant difference between the groups, likely due to the high variability observed in the RT plus epicatechin group. In this investigation, the absolute plasma myostatin levels (assessed using ELISA, R&D Systems, Minneapolis, MN, USA) did not appear to be consistent with observational research assessing myostatin levels in different populations; for example, the baseline level of ~1300 ng/mL was much higher than the reference ranges discussed in Topic 3.

#### 4.2.5. Vitamin D

Although vitamin D is recognized as the main hormone of bone metabolism, its active form (i.e., calcitriol) along with the vitamin D receptor modulate skeletal muscle function by genomic and hormonal effects [84,85], indicating potential effects of vitamin D supplementation on skeletal muscle strength and growth, the clinical magnitude of which is modest but deserves attention, especially for individuals at high risk for vitamin D deficiency [84,86]. Research evaluating the effect of vitamin D on myostatin are limited. In a study that investigated the effects of 20 μg/d (n = 25) of oral calcifediol or 30 μg/day (n = 25) for 180 days in postmenopausal women, serum myostatin levels did not change significantly in either group [87].

### 4.3. General Considerations

Although a drop in myostatin levels, based on mechanistic aspects, would be expected when supplementing with protein, amino acids and derivatives, compelling clinical studies show similar effects compared to placebo, with some exceptions favoring intervention or placebo. Therefore, there is no evidence that supplementing with protein, amino acids, and derivatives perform important actions in modulating myostatin levels.

There are only a handful of studies regarding non-protein supplements (functional foods, herbal extracts, and supplementation with micronutrients and antioxidants), which support neutral effects on myostatin levels.

Taken together, the effects of supplements and functional foods on myostatin levels (skeletal muscle mRNA, serum, or plasma) are summarized in Table 2.

## 5. Physical Exercise

### 5.1. Resistance Training

In humans, introducing RT can downregulate muscle myostatin mRNA in untrained young and older adults [88,89,90] for both men and women. In a study including untrained young and older men (n = 7) and women (n = 8), 9 week RT of unilateral knee extension decreased myostatin mRNA expression from vastus lateralis by 37% in all subjects [90]. In addition, even a single bout of RT can exert this effect [89], so much so that in a study composed of heathy untrained older (15 women, 14 men) and younger (16 women, 21 men) adults, 24-h acute response to the first RT loading bout decreased myostatin mRNA expression by 44%, maintaining a 52% suppression relative to baseline after 16 weeks of RT primarily focused on knee extensor training [91].

The detraining process, in turn, can markedly increase myostatin expression. Following 90 days of RT in young untrained males, myostatin mRNA expression significantly increased by 56%, 79%, 107%, and 76% after 10, 30, 60, and 90 days in the detraining period [92].

### 5.2. Concurrent Training

Concurrent training can reduce myostatin levels irrespective of the order. In older men with sarcopenia (n = 30) randomized to endurance training followed by RT or RT followed by endurance training, serum myostatin levels reduced by 308 pg/mL and 294 pg/mL, respectively, while those randomized to control group maintained their myostatin levels [93].

### 5.3. Clinical Populations

Interestingly, another study did not find changes in myostatin expression in healthy volunteers using two weeks of limb immobilization, but one day following the removal of the limb cast and the first bout of exercise training, myostatin expression was significantly down-regulated by 48% compared to with post-immobilization [94]. In this study, however, continuation of rehabilitation training for 6 weeks did not further reduce myostatin expression, but at least maintained expression lower than post-immobilization phase.

Intriguingly, total myostatin levels did not differ statistically between tetraplegic patients and bodybuilders (~26 vs. ~33 ng/mL for tetraplegic and bodybuilders, respectively; *p*>0.05) [95]. Such data are of pivotal importance because they compare the myostatin status in a population that suffers muscle atrophy with another that encompasses muscularity performance at the athletic level. However, it must be noted that the bodybuilders group had higher myostatin propeptide levels compared to healthy untrained young men controls (~81 vs. ~35 pg/mL) and the tetraplegic patients (~81 vs. ~31 pg/mL), whose biomarker is a potent myostatin inhibitor [95].

At last, a 10 week RCT consisting of men with chronic obstructive pulmonary disease submitted to weekly testosterone injection alone, weekly testosterone injection plus RT, weekly placebo injection, or weekly placebo injection plus RT, myostatin mRNA expression was unchanged between groups [96].

### 5.4. General Considerations

Although certain downregulation of muscle myostatin mRNA along with decreased circulating myostatin levels can be expected in interventions with RT, there are no uniform changes in myostatin status upon exercise programs and neutral effects can be expected in some instances.

Adherence to RT programs is the cornerstone of muscle hypertrophy regardless of putative myostatin-mediated modulation. Currently, personalized high-volume techniques are the most variable in RT programs for maximizing muscle gains [97,98].

## 6. Pharmacological Agents

### 6.1. Specific Myostatin-Blocking Drugs

The approaches to blocking myostatin activity show promise for clinical application. Several myostatin-blocking drugs have evoked promising clinical approaches, such as myostatin propeptide derivatives (e.g., wild-type myostatin propeptide, recombinant AAV8 vector, and plasmid-mediated deliveries), and other myostatin inhibitors, such as follistatin, follistatin-related gene, and G protein-coupled receptor-associated sorting protein 1; however, most of the scientific background remains restricted to animal models (laboratory mice and livestock) [99,100].

### 6.2. Testosterone

Myostatin status in those receiving hormone replacement therapy, particularly with anabolic hormones, is relevant as a means of understanding myostatin’s variation in the hormone-deficient state and after the hormone correction. Testosterone replacement therapy can perhaps be considered the most common androgen hormone treatment in males [101,102], whose anabolic profile merits discussion in the field of myostatin.

In men treated with graded doses of testosterone (25, 50, 125, 300, or 600 mg testosterone enanthate intramuscularly injected weekly), serum myostatin levels were significantly higher at day 56 than baseline in young and older men, but the effect was transient and returned to a similar baseline state at day 140 of treatment [32]. Thus, myostatin can be a counter-regulatory hormone upon testosterone administration, in which increasing circulating myostatin levels may be a key factor in restraining unlimited skeletal muscle growth. However, it is important to note that the individuals had normal testosterone levels and the graded doses encompass low to supraphysiological doses of testosterone, hence, limiting a direct translation into the male hypogonadism treatment.

A study involving men with hypogonadotropic hypogonadism, in turn, found myostatin muscle expression decreased by 29 ± 12% after 22 weeks of traditional testosterone replacement therapy (250 mg testosterone every 2 weeks), but serum myostatin did not change [103]. Interestingly, this study observed that men with hypogonadotropic hypogonadism had approximately half the serum myostatin concentrations compared with eugonadal men (2.23 ± 0.23 vs 4.0 ± 0.5 ng/mL).

In addition, testosterone suppression does not necessarily affect myostatin levels. In a clinical trial consisting of 22 young men randomized to an anti-androgen drug (3.6 mg monthly GnRH analogue goserelin) or placebo for 12 weeks, accompanied by 8 weeks of RT starting at week 4, myostatin mRNA expression from vastus lateralis decreased in both groups after the RT period without between-group differences [104].

### 6.3. Growth Hormone

Although growth hormone (GH) use is commonly targeted at bone metabolism (e.g., formation and resorption) [105], the GH and IGF-1 axis can stimulate skeletal muscle hypertrophy signaling pathways while suppressing atrophy related genes [60,106,107].

Skeletal muscle myostatin mRNA expression and plasma myostatin levels are 3 and 1.5-fold higher in GH-deficient participants than in healthy individuals (3.2 ± 2.7 μg/L versus 2.1 ± 1.9 μg/L for plasma levels) [108]. In adult patients with GH-deficient hypopituitary, myostatin mRNA expression was significantly reduced by 31 ± 9% after a 6 month GH replacement (Saizen; 5 μg/kg·night injected subcutaneously), whose levels were maintained after 12 and 18 months of GH treatment [109].

To the best of our knowledge, the use of GH and its effects on myostatin levels are geared towards GH-deficient participants, and, therefore, little is known about the effect of supraphysiological use in this regard.

### 6.4. General Considerations

Notwithstanding the development of myostatin-blocking drugs and results based on phase I and II RCTs for muscle wasting disorders and diseases [34,110], it is warranted to wait for the consolidation of phase III RCTs and medical scoring guidelines to draw better clinical conclusions.

In addition, thus far, the benefits of hormone replacement therapy with anabolic potentials, such as treatment with testosterone and GH injections, should not be based on putative inhibition of myostatin levels.

## 7. Take-Home Messages

The effects of dietary supplements, nutraceutical agents, and physical exercise on myostatin mRNA expression in skeletal muscle and serum myostatin levels are not uniform, and there may be reductions, increases, or neutral effects. The large amount of research using RT protocols shows that supplements or functional foods do not clearly outperform placebo for modulating myostatin levels. In some cases, laboratory changes occurred only in the intervention group or only in the placebo group, but all changes are of dubious clinical magnitude for the outcome of muscle mass accretion.

Thus, despite some biological hope in using supplements or certain functional foods to decrease myostatin levels, caution must be exercised not to propagate the hype of the food supplement market, select health professionals, and laypeople. Importantly, putative changes in myostatin levels represent only one factor influencing the complex processes of muscle accretion and atrophy and are not necessarily the key factor to consider for the primary aims of muscle growth and strength development.

Whereas proponents of some supplements and other nature-based interventions tout the ability to modulate myostatin levels, based on limited evidence or anecdote, the main clinical recommendations for muscle growth and strength remains an appropriate RT program along with a personalized dietary plan, particularly including 1.6–2.0 g/kg/d total daily protein regardless of source [111,112,113], individually appropriate energy and nutrient intake, and sleep habits [114,115].

## Figures and Tables

**Table 1 metabolites-12-01146-t001:** Serum myostatin levels in different populations.

Population	Serum Myostatin Levels	Reference
Young and older male (n = 1121)	30.5 ± 9.5 ng/mL for 20 to 30 yr (n = 76)	[30]
	26.7 ± 10.6 ng/mL for >30 to 40 yr (n = 69)	
	28.3 ± 10.1 ng/mL for >40 to 50 yr (n = 88)	
	32.7 ± 10.1 ng/mL for >50 to 60 yr (n = 91)	
	30.6 ± 11.9 ng/mL for >60 to 70 yr (n = 314)	
	28.4 ± 12.3 ng/mL for >70 to 80 yr (n = 345)	
	28.9 ± 10.4 ng/mL for >80 yr (n = 121)	
Older individuals (n = 1053)	3.7 ± 1.2 ng/mL for men	[31]
	3.2 ± 1.1 ng/mL for women	
Caucasian women (in perimenopause or postmenopause) (n = 300)	6.58 ± 3.59 ng/mL	[33]
Menstruating women (n = 33), Naturally menopausal (n = 24), and Surgically menopausal women (n = 37)	7.0 ± 2.7 ng/dL for menstruating women	[32]
	6.7 ± 2.8 ng/mL for naturally menopausal	
	6.7 ± 2.7 surgically menopausal women	
Kidney transplantation recipients (n = 84)	6.99 (5.82–8.32) ng/mL	[35]
Heart failure (n = 41)	18.7 ± 7.4 ng/mL	[36]
Metabolic syndrome (n = 204)	7.39 ± 3.46 ng/mL	[37]
Type 2 diabetes (n = 246)	7.82 ± 3.85 ng/mL	[37]
Dermatomyositis and polymyositis (n = 50)	16.9 ± 12.1 ng/mL	[38]
Hemodialysis (n = 140)	40.18 ± 8.36 ng/mL	[39]
Hemodialysis (n = 204)	2573 (1662; 3703) pg/mL	[40]
Hemodialysis (n = 60)	25.7 ± 12.8 μg/mL	[41]
Peritoneal dialysis (n = 69)	7.59 ± 3.37 ng/mL	[42]
Liver cirrhosis (108 men and 90 women)	3419.6 pg/mL (578.4–12897.7 pg/mL) for men	[43]
	2662.4 pg/mL (710.4–8782.0 pg/mL) for women	
Liver cirrhosis (n = 115)	1.14 (0.57–2.19) ng/mL	[44]
Advanced chronic liver disease (198 men and 90 women)	1959.4 pg/mL (1082.8, 3914.8) for men and 1790.1 pg/mL (914.1, 3158.7) for women	[45]
Alcoholic hepatitis (n = 131)	1.58 ng/mL (IQR 0.73, 3.17) for men	[46]
	0.84 ng/mL (IQR 0.56, 1.76) for women	
Heavy drinkers (n = 124)	3.06 ng/mL (IQR 2.25, 4.08) for men	[46]
	2.01 ng/mL (IQR 1.66, 3.07) for women	
Children with T1DM (n = 87)	23.60 ± 7.70 ng/mL	[47]
Healthy children (n = 75)	16.74 ± 6.95 ng/mL	[47]
Duchenne muscular dystrophy (n = 74)	1.1 ± 0.8 ng/mL	[48]
Chronic obstructive pulmonary disease (n = 70)	11.85 ± 4.01 ng/mL	[49]

**Table 2 metabolites-12-01146-t002:** Effects of supplements and functional foods on myostatin levels (skeletal muscle mRNA, serum, or plasma) based on RCTs.

Reference	Participants	Duration	Dietary Intervention	Myostatin Levels
*Proteins, amino acids, and derivatives*				
Hulmi et al. (2008) [67]	18 trained middle-aged to older men	Acute and chronic (21 week)	15 g of whey protein both before and after exercise	↓myostatin mRNA levels at 48 h assessment postexercise for placebo
Hulmi et al. (2009) [68]	31 untrained young men	Acute and chronic (21 week)	15 g of whey protein both before and after exercise	↓31% myostatin mRNA expression only in the placebo group at 1 h post-exercise
Dalbo et al. (2013) [69]	10 untrained college-aged men (crossover)	Acute	Whey protein isolate (25 g), maltodextrin (25 g), or placebo 30 min prior to RT	↓myostatin mRNA expression reduced in all conditions (−29.4% for placebo, −24.7% for protein, and −3.4% for carbohydrate) 6 h after RT
Amasene et al. (2022) [23]	41 hospitalized older individuals	12 week	Whey protein (20 g) + leucine (3 g)	↔
Dirks et al. (2014) [58]	23 older individuals on immobilization of one knee	5 d	Whey protein (20.7 g protein)	↔
Constantin et al. (2013) [59]	59 patients with COPD and 21 healthy controls	8 week	Whey protein concentrate + glucose polymer carbohydrate (19 g protein, 49 g carbohydrates)	↔
Pourabbas et al. (2021) [24]	30 resistance-trained young men	6 week	High-protein milk (30 g of protein, 6 g from whey and 24 g from casein)	↓serum myostatin levels compared to placebo
Wette et al. (2021) [64]	7 men on RT (crossover)	Acute	Postexercise meal milk protein (600 mL chocolate milk and 85 g muesli bar, 102 g carbohydrate, 34 g protein, and 22 g fat)	↔
Li et al. (2015) [51]	41 young men	Acute	Peri-exercise supplementation of carbohydrates (1.5 g/kg), carbohydrates + BCAA (120 mg/kg BCAA), or carbohydrates + leucine (120 mg/kg)	myostatin mRNA expression over 360 min postexercise was higher for those who supplemented with carbohydrates (1.00 ± 0.09) and carbohydrates + BCAA (1.05 ± 0.08) than carbohydrates + leucine (0.92 ± 0.07) or placebo (0.90 ± 0.05)
Bagheri et al. (2021b) [53]	20 postmenopausal women on RT	8 week	BCAA (9 g/d)	↔
Tsien et al. (2015) [56]	6 men with cirrhosis and 8 healthy controls	Acute	BCAA (7.5 g of leucine, 3.75 g of isoleucine, and 3.75 g of valine)	↔
Olveira et al. (2016) [57]	28 patients with bronchiectasis	12 week	HMB-enriched protein supplementation (18 g protein, 1.5 g HMB)	↔
Saremi et al. (2010) [54]	24 men	8 week	Creatine (0.3 g/kg/d BW at week 1 and 0.05 g/kg BW/d for the rest)	↓plasma myostatin levels in creatine and placebo groups
Deldicque et al. (2008) [55]	9 young men on RT	Acute (measures after 5 d of creatine loading)	Creatine (21 g/d; 7 g 3 x/d)	↔
Sire et al. (2019) [70]	20 patients submitted to hip replacement	8 week	Amino acids (4 g 2 x/d)	↓serum myostatin levels for amino acid (from 1.2 ± 0.2 to 0.9 ± 0.3 ng/mL) and placebo (from 1.3 ± 0.3 to 1.1 ± 0.4 ng/mL) groups
Paoli et al. (2015) [22]	18 active young men without experience with RT	8 week	High-protein diet (1.8 g protein/kg BW/d) vs. normal-protein diet (0.85 g protein/kg BW/d)	↑plasma myostatin levels after RT session (pretraining and posttraining levels: from 3.66 ± 1.42 to 12.0 ± 2.5 ng/mL) in the high-protein group
Bagheri et al. (2020) [52]	30 young men	12 week	Whole eggs (3 units) vs. egg whites (6 units)	↔
*Non-protein supplements*				
Willoughby (2004) [75]	22 untrained health men	12 week	*Cystoseira canariensis* (1200 mg/d)	↔
García-Merino et al. (2020) [81]	44 training endurance athletes (men)	10 week	Cocoa (5 g cocoa powder, 425 mg flavonoids)	↔
Mafi et al. (2018) [83]	62 older individuals	8 week	Epicatechin (1 mg/kg BW/d	↓49% in plasma myostatin levels in the RT plus epicatechin group than in the RT group
Gonnelli et al. (2021) [87]	50 postmenopausal women	180 d	Calcifediol (20 or 30 μg/d)	↔
Bagheri et al. (2021a) [79]	40 male wrestlers	12 d	Spirulina (3 g/d)	↓serum myostatin levels by 0.1 ng/mL

Notes: BCAA—branched-chain amino acids; BW— body weight; COPD—chronic obstructive pulmonary disease; HMB—beta-hydroxy-beta-methylbutyrate; RT—resistance training.

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
