# Peer review of "The Effects of Dietary Supplements, Nutraceutical Agents, and Physical Exercise on Myostatin Levels: Hope or Hype?"

_metabolites, 2022, doi:10.3390/metabo12111146_

Round 1
Reviewer 1 Report
Overall, this review is an informative and useful for concerns about therapeutic potential of myostatin.
Main concerns;
1) p.1 L.40-44
To avoid giving the impression that myostatin never have been considered to a therapy for muscular dystrophy diseases, the authors should comment on clinical trials (e.g., Leung DG, et al, 2015; 10.1002/mus.24569) and attentions as a potential target to treat the metabolic syndrome (e.g., Palsgaard J, et al., 2009; 10.1371/journal.pone.0006575).
2) p.2 L.58-70
Previously findings should be taken into consideration while discussing the mechanisms of myostatin. For example, previously studies said how the precursor of myostatin is processed after translation, how myostatin activates Smad complex, or how it negatively regulates muscle regeneration.
3) p.5 L.200
To clarify the contribution of BCAA to muscle protein synthesis, it would be appropriate to explain that BCAA activates an anabolic signal in skeletal muscle at the beginning of the 4.1.4. Amino acids section.
4) p.11 L.1 and p.11 L.45
The authors present a number of intriguing data, but it is difficult to follow the relationship between observations and authors’ conclusions in subtopics. It would be helpful to include the authors’ suggestion or argument in the physical exercise and pharmacological agents subsection.
Minor concerns;
1) p.3 L.101-102
The position of the reference number is not correct.
2) p.6 L. 240
Both P and p are used; “p” in this sentence is lowercase letter.
3) p. 7 L. 291-293
This sentence is repeated twice.
4) p.11 L.2-5
Correct the position of Notes.
5) p. 11 L. 48
The abbreviation “MSTN” is not defined.
6) p.12 L.52
Is it not so much the scientific background as the scientific experiment?
7) p. 12 L.54-56
Myostatin status in hormone replacement is discussed in the specific myostatin-blocking drugs section. Since this discussion is related to Testosterone, it should be discussed in the Testosterone section.
8) p. 12 L.101
There is a no start of parenthesis, although there is an end of it.
9) Table 2
It would be helpful to indicate “Duration” represents the duration of RT.
Author Response
Reviewer 1 minor
Overall, this review is an informative and useful for concerns about therapeutic potential of myostatin.
Answer: Dear reviewer, thank you very much for your suggestions. Undeniably, our paper has improved after your observations and the corresponding changes.
Main concerns;
1) p.1 L.40-44
To avoid giving the impression that myostatin never have been considered to a therapy for muscular dystrophy diseases, the authors should comment on clinical trials (e.g., Leung DG, et al, 2015; 10.1002/mus.24569) and attentions as a potential target to treat the metabolic syndrome (e.g., Palsgaard J, et al., 2009; 10.1371/journal.pone.0006575).
Answer: We now cite some types of muscular dystrophy diseases and used the reference you suggested. Please, check “…many myostatin-blocking agents have gained attention in agriculture applications and in the management of muscle diseases and disorders (6)—e.g., injuries, sarcopenia, wasting/cachexia, Duchenne type muscular dystrophy, Becker muscular dystrophy, Facioscapulohumeral muscular dystrophy, etc. (7, 8)…”.
Regarding metabolic syndrome, we used the reference you suggested in concert with other ones, but focused on the management of glucose/insulin levels to be more specific. Please, check: “In addition, experimental research has shed light on myostatin inhibition in muscle to improve insulin resistance by enhancing glucose uptake (9-11).”
2) p.2 L.58-70
Previously findings should be taken into consideration while discussing the mechanisms of myostatin. For example, previously studies said how the precursor of myostatin is processed after translation, how myostatin activates Smad complex, or how it negatively regulates muscle regeneration.
Answer: We rewrote Topic 2 (“Mechanisms”) adding details about the relationship between myostatin, Smad complex, and related negative regulation in muscle mass.
3) p.5 L.200
To clarify the contribution of BCAA to muscle protein synthesis, it would be appropriate to explain that BCAA activates an anabolic signal in skeletal muscle at the beginning of the 4.1.4. Amino acids section.
Answer: We clarified the contribution of BCAA to muscle protein synthesis in this section. Please, check: “Proteins, amino acids, and derivatives Adequate protein intake is fundamental to optimize muscle hypertrophy and minimize muscle catabolism, in which essential amino acids, e.g., branched-chain amino acids (BCAAs), increase mTOR phosphorylation and sequential activation of 70-kD S6 protein kinase and the eukaryotic initiation factor 4E-binding protein 1 (49).
4) p.11 L.1 and p.11 L.45
The authors present a number of intriguing data, but it is difficult to follow the relationship between observations and authors’ conclusions in subtopics. It would be helpful to include the authors’ suggestion or argument in the physical exercise and pharmacological agents subsection.
Answer: At the end of the topics about physical exercise and pharmacological agents, we added a subtopic named “General considerations” to summarize the data.
Minor concerns;
1) p.3 L.101-102
The position of the reference number is not correct.
Answer: We corrected the position of the reference number.
2) p.6 L. 240
Both P and p are used; “p” in this sentence is lowercase letter.
Answer: We standardized all p-values to lowercase and italics after checking the journal guidelines.
3) p. 7 L. 291-293
This sentence is repeated twice.
Answer: We corrected this part.
4) p.11 L.2-5
Correct the position of Notes.
Answer: We corrected the position of Notes.
5) p. 11 L. 48
The abbreviation “MSTN” is not defined.
Answer: We replaced MSTN with myostatin.
6) p.12 L.52
Is it not so much the scientific background as the scientific experiment?
Answer: We added the term “most” to alleviate the assertion. So, the current phrase is “…but most of the scientific background remains restricted to animal models (laboratory mice and livestock)”.
7) p. 12 L.54-56
Myostatin status in hormone replacement is discussed in the specific myostatin-blocking drugs section. Since this discussion is related to Testosterone, it should be discussed in the Testosterone section.
Answer: We have put this part in the testosterone section.
8) p. 12 L.101
There is a no start of parenthesis, although there is an end of it.
Answer: We have removed the parenthesis.
9) Table 2
It would be helpful to indicate “Duration” represents the duration of RT.
Answer: Since Table 2 covers both clinical populations without physical exercise intervention, and participants who underwent RT, we cannot indicate that duration represents the length of RT for all studies. However, the duration cited for all the studies with RT interventions represents the length of the exercise protocol.

Reviewer 2 Report
1. Line no 37: many myostatin-blocking biotechnologies……………….it is not correct, maybe the author wants to explain about many myostatin-blocking agents?
2. Line no 37- 39: This line is very confusing. Try to rewrite the sentence.
3. Line no 39: some forms of muscular dystrophy……it is not correct, if ok then please write the name of different forms. In same, author mentioned injury……..injury is not a disease it may be a condition/ disorder.
4. Line no 40: myostatin is seemingly not considered a therapeutic target for muscular diseases. It is a well-known target for Skeletal muscle improvement and many studies are reported to inhibit myostatin for Skeletal muscle improvement.
5. In the same line: what is per se?
6. Line no 45-46: It is noteworthy that, to date, there is no consensus on the use of myostatin-blocking drugs or non-pharmacological approaches in real-world scenarios. Please add the proper reference for this sentence. There are few drugs under clinical trials so write the above sentence by considering these drugs.
7. Overall, the introduction is not sufficient it can be improved.
8. Mechanism is also lacking several things related with interaction of myostatin with its receptor ActR2B.
9. The myostatin levels was found lower in Korean population and Korean people have a longer average life. Is there any relation between Myostatin level and age? If yes, then please write a conclusion in the same paragraph.
10. The text related to myostatin blocking in the introduction and subheading 6.1 is conflicting with each other. Please revise the sentences in the introduction as “there is no consensus on the use of myostatin-blocking 45 drugs or non-pharmacological approaches in real-world scenarios”.
11. Myostatin is a protein, not an enzyme. Is it ok to inhibit Myostatin or ActR2B which is a receptor of myostatin, so blocking of ActVR2B is good or inhibition of Myostatin? Please discuss this point in one paragraph. It may help to gain more attention to this manuscript.
Author Response
Review 2 major
Answer: Dear reviewer, thank you very much for your comments. In fact, our manuscript has reached a better state after your perusal and the corresponding changes.
- Line no 37: many myostatin-blocking biotechnologies……………….it is not correct, maybe the author wants to explain about many myostatin-blocking agents?
Answer: We replaced myostatin-blocking biotechnologies with myostatin-blocking agents.
- Line no 37- 39: This line is very confusing. Try to rewrite the sentence.
Answer: We rewrote this sentence.
- Line no 39: some forms of muscular dystrophy……it is not correct, if ok then please write the name of different forms. In same, author mentioned injury……..injury is not a disease it may be a condition/ disorder.
Answer: We rewrote this sentence accordingly. Please, check: Since myostatin was cloned in 1997 (5), many myostatin-blocking agents have gained attention in agriculture applications and in the management of muscle diseases and disorders (6)—e.g., injuries, sarcopenia, wasting/cachexia, Duchenne type muscular dystrophy, Becker muscular dystrophy, Facioscapulohumeral muscular dystrophy, etc. (7, 8).
- Line no 40: myostatin is seemingly not considered a therapeutic target for muscular diseases. It is a well-known target for Skeletal muscle improvement and many studies are reported to inhibit myostatin for Skeletal muscle improvement.
Answer: We agree that myostatin is a well-known target for skeletal muscle improvement and many studies are reported to inhibit myostatin for skeletal muscle improvement. However, in this sentence, we are considering the magnitude of the high-grade evidence provided by guidelines, thinking about the current decision-making by practitioners.
That said, we rewrote this sentence trying to maintain the importance of emerging myostatin-blocking drugs, but mentioning that apparently medical guidelines still do not rate myostatin by itself as a therapeutic target. Below is the new sentence:
“Despite emerging myostatin-blocking drugs, myostatin alone is apparently not considered by guidelines as a therapeutic target for muscle diseases in the same manner as low-density lipoprotein-cholesterol for hypercholesterolemia and related heart diseases, blood pressure for hypertension, fasting blood glucose for diabetes, or blood total testosterone for male hypogonadism”
- In the same line: what is per se?
Answer: Per se was used as a synonym for “by itself” or “alone”. We have replaced “per se” with “alone” when rewriting the sentence, as mentioned above.
- Line no 45-46: It is noteworthy that, to date, there is no consensus on the use of myostatin-blocking drugs or non-pharmacological approaches in real-world scenarios. Please add the proper reference for this sentence. There are few drugs under clinical trials so write the above sentence by considering these drugs.
Answer: We have deleted this sentence and created a new one based on your suggestion and after checking the https://clinicaltrials.gov/. Please, check: “To date, there are ongoing randomized clinical trials (RCTs) investigating the effects of myostatin-blocking drugs (15-17).”
- Overall, the introduction is not sufficient it can be improved.
Answer: We have made amendments to the introduction in an attempt to improve it. However, we also sought to keep the introduction succinct due to the current length of the manuscript.
- Mechanism is also lacking several things related with interaction of myostatin with its receptor ActR2B.
Answer: In Topic 2 (“Mechanisms”), we added more details about the relationship between myostatin and its receptors.
- The myostatin levels was found lower in Korean population and Korean people have a longer average life. Is there any relation between Myostatin level and age? If yes, then please write a conclusion in the same paragraph.
Answer: We added the negative relationship between age and myostatin level in this part. Please, check: “Although a certain decline in serum myostatin levels can be expected for older subjects, seemingly due to physiological actions to partially counteract age-related muscle wasting (34), in this Korean study (31),…”
Regarding the conclusion, we used the last paragraph of topic 3 to summarize the data. Also, we added a last sentence to endorse the importance of establishing reference ranges for myostatin levels “However, further research is crucial to propose specific reference ranges for circulating myostatin levels.”
- The text related to myostatin blocking in the introduction and subheading 6.1 is conflicting with each other. Please revise the sentences in the introduction as “there is no consensus on the use of myostatin-blocking 45 drugs or non-pharmacological approaches in real-world scenarios”.
Answer: As mentioned in a previous answer, we removed this sentence in the introduction, and updated it based on ongoing studies registered on https://clinicaltrials.gov/ (references 15,16, and 17).
- Myostatin is a protein, not an enzyme. Is it ok to inhibit Myostatin or ActR2B which is a receptor of myostatin, so blocking of ActVR2B is good or inhibition of Myostatin? Please discuss this point in one paragraph. It may help to gain more attention to this manuscript.
Answer: We agree that myostatin is a protein, not an enzyme. Correspondingly, we searched the term “enzyme” to verify if this term was not cited to classify the myostatin, and we observed that we cited the term “enzyme” two times (14-kDa ubiquitin-conjugating enzyme E2 and sandwich enzyme immunoassay kit) only for referring to a specific pathway whereby myostatin acts and to an analytical technic.
We also are in agreement that both blockers of Myostatin and its receptor deserve an interest. Thus, we cited more details about this receptor and added more background to create a link to support the proposals for blocking both myostatin and its main receptor. Please, check Topic 2 (“Mechanism”).

Round 2
Reviewer 2 Report
The manuscript is improved